# LTCC-Integrated Dielectric Resonant Antenna Array for 5G Applications

**DOI:** 10.3390/s21113801

**Published:** 2021-05-31

**Authors:** Mohsen Niayesh, Ammar Kouki

**Affiliations:** Electrical Engineering Department, Ecole de Technologie Superieure (ETS), Montreal, QC H3C 1K3, Canada; ammar.kouki@etsmtl.ca

**Keywords:** millimeter wave (mm-Wave), dielectric resonant antenna (DRA), fifth generation (5G), array antenna, corporate feeding network, multi-layered technology, low-temperature cofired ceramics (LTCC)

## Abstract

A millimeter-wave dielectric resonator antenna array with an integrated feeding network and a novel alignment superstrate in Low Temperature Cofired Ceramics (LTCC) technology is presented. The antenna array consists of 16 cylindrical DR antenna (CDRA) elements operating at 28 GHz for mm-Wave 5G applications. The array is fed by an inverted microstrip corporate feeding network designed and built in the same LTCC stack as the resonators. A grooved and grounded superstrate is introduced to facilitate the alignment of the individual array elements while enhancing the overall performance of the antenna array. The performance of the proposed stack is evaluated numerically and compared with measured data. Measured results show an impedance bandwidth of 9.81% at 28.72 GHz with a maximum realized gain of 15.68 dBi and an efficiency of 88%, and are in excellent agreement with simulations.

## 1. Introduction

Millimeter-wave (mm-Wave) frequency bands have been selected for 5G applications as a response to the demand for higher data transmission rates in wireless communications. One of the challenges in these bands is the increased link loss due to the reduced wavelength and atmospheric absorption. Consequently, mm-Wave antenna systems are required to provide high gain and high efficiency [1,2,3,4,5,6,7,8,9,10,11]. Various approaches and implementations of radiating elements have been proposed for the mm-wave frequency bands, including flexible antennas [6,7], 3D printed antennas [1,11], microstrip patches and slots [2,4,8,9,10] and conventional DRAs on printed circuit board (PCB) technology [3,5,9]. Printed antennas, i.e., patches and slots, are intrinsically narrowband and suffer from conductor losses as well as the likely excitation of surface waves, both of which lead to lower efficiency. Different approaches have been proposed to increase bandwidth, such as using multiple layer stacks, perfect magnetic conductors (PMC) and defective grounds [12,13,14]. Others have focused on reducing losses associated with surface waves [15,16]. Still, the bandwidth and efficiency performance of these antennas remain limited. On the other hand, dielectric resonant antennas (DRAs) can provide high efficiency and relatively wide bandwidth [17]. Indeed, DRAs have witnessed significant developments in the last decade and have become a promising candidate for higher frequency applications, such as mm-Wave 5G due to their compact size, high power handling capability and potential broadband response [18]. Compared to planar (multilayer) antennas, DRAs can offer better bandwidth and efficiency but are generally costlier and require a more involved fabrication process [17].

DRAs consist of volumetric dielectric structures that are excited via an electromagnetic coupling mechanism, such as a microstrip line, an aperture in a conducting plane or a feeding probe [3,19,20]. Many efforts have been deployed to improve the bandwidth and gain of a single DRA element such as Mrnka and Raida [21] and Ta and Park [22]. One of the ways to increase the gain is exciting higher-order resonator modes [21,23]. These techniques have led to gain, efficiency and bandwidth performance that are better than the printed antennas. However, using DRAs with standard PCB fabrication technology can be sensitive to tolerances during the machining of the resonators themselves and, more frequently, to tolerances when aligning and shaping the resonators with the feeding structure on the PCB. This becomes even more critical at higher frequencies where high alignment precision and tight size tolerances are required to avoid performance degradation. For instance, in Abdallah et al. [20], due to the fabrication process, a frequency shift of 1.5 GHz and a reduction in gain of 2.5 dB at the desired resonant frequency of the designed DRA were reported and attributed to fabrication and assembly. To address DRA cost, machining precision, and alignment tolerances, while maintaining good performance, the use of Low Temperature Cofired Ceramics technology (LTCC) for DRA fabrication and integration can provide a viable solution. Indeed, LTCC facilitates the fabrication of DRA-based antenna systems and their direct integration with other front-end circuits, e.g., [24,25], in a single process. An initial demonstration of this concept was proposed by Niayesh and Kouki [26] where a single DRA element was fabricated in LTCC for operation at the X-band. Although good results were achieved, the proposed design was found not to be suitable for use at higher frequencies because it was based on probe feeding, which presents fabrication and alignment challenges in an array context at such frequencies.

In this paper, an efficient LTCC-based dielectric resonator antenna array with aperture feeding suitable for mm-wave frequencies and 5G applications is presented. The array is fabricated in a single LTCC process with a new custom layer stack that provides highly precise dimensioning and alignment. The proposed structure provides a wide bandwidth, stable gain over the operating frequency band and high radiation efficiency. The organization of this paper is as follows. In Section 2 the design of the proposed single element DRA is discussed. In Section 3, the antenna array design procedure, including the antenna array configuration and feeding network, is presented and with numerical and measurement results of the proposed array. Section 4 concludes this paper.

## 2. Single Element DRA Design

In this section, the design of a single element Cylindrical Dielectric Resonator Antennas (CDRA) using LTCC technology is presented. First, a conventional design approach, similar to that with standard PCBs, is used, whereby the DR and the substrate are both made out the same ceramic material but aligned manually with limited precision. Second, a new design is proposed whereby the alignment precision is addressed through the introduction of a grooved superstrate ceramic layer. In both designs, the LTCC material used is the Ferro A6M green tape with *ε_r_* = 5.8 and loss tangent of 0.001. All conductors are made of silver (σ = 6.1 × 10^7^ S/m) with a thickness of 8 µm. The basic theory behind the design of DRAs is well developed by Petosa [27] and consists mainly of dimensioning the resonator, usually in a cylindrical shape, although other shapes have been proposed [28,29,30], such that a given mode is excited, usually a hybrid HE or an EH mode.

### 2.1. Conventional CDRA Design in LTCC

The conventional single element CDRA design consists of a cylindrical resonator, typically a high-dielectric ceramic [17,28], placed on top of the ground plane of a PCB substrate, typically made of organic material [17], with one of many different feeding mechanisms [17,28,31]. Here, we propose to follow the same procedure as the conventional design method, but using the LTCCs as both the ceramic of the cylindrical resonator and the grounded substrate on which it is placed. We further choose a feeding mechanism consisting of an inverted microstrip line that couples to the resonator through an aperture, of dogbone shape in the ground plane as illustrated by the side view in Figure 1 and the top schematic view in Figure 2. The design parameters consist of the substrate’s height, *h_sub_*, the resonator’s radius, *r_DR_*, and height, *h_DR_*, the feeding line’s width, *w_f_*, the dogbone slot’s dimensions, *W*_s_, *L_srm_*, and *W_srm_*, and the off-set of the slot from the end of the feeding line, *L*_of_.

In order to achieve a large bandwidth and a broad radiation pattern, we first dimensioned the DR such that it would resonate in the HEM_11__δ_ mode at 29.5 GHz, the center frequency of the band, with *ε_r_* = 5.8. Based on Petosa [27], we chose *r_DR_* and found it to be 2.5 mm. Next, we the closed form expression for the resonant frequency, *f*_r_ [32]:(1)$$f_{r}=\frac{c}{2\pi r_{DR}{\sqrt{\varepsilon }}_{r}}\left[1.71+\left(\frac{r_{DR}}{h_{DR}}\right)+0.1578{\left(\frac{r_{DR}}{h_{DR}}\right)}^{2}\right]$$

We found the value of *h_DR_* = 1.02 mm. These values, *r_DR_* = 2.5 mm and *h_DR_* = 1.02, are used as the initial dimensions for the DR and were then optimized numerically by using the eigen-mode solver in HFSS to ensure that the HEM_11δ_ mode resonated at 29.5 GHz. This optimization yields a DR radius of 1.95 mm and a height of 1.45 mm with a mode distribution of the magnetic field as illustrated in Figure 3. The substrate is made of four 5-mil LTCC sheets to ensure mechanical robustness. This leads to a fired substrate thickness, *h_sub_*, of 0.5 mm and a 50 Ω inverted microstrip line width, *w_f_*, of 1.75 mm. The slot shape, dimensions and position were optimized to excite the HEM_11δ_ mode and performance. The slot width, *Ws*, controls the level of coupling between the microstrip and the DR while the proper values of *L_srm_* and *W_srm_* ensured the best uniformity of the mode distribution [33]. The offset of the slot from the end of the microstrip line, *L_off_*, was used to tune the matching. The optimal values of all these parameters were summarized in Table 1 and were obtained by optimization in HFSS that maximized the gain and bandwidth and minimized the return loss.

Figure 4 shows the magnetic field distribution in the DR with the optimized feeding structure at the center frequency of 29.5 GHz, which is found to be very close to the desired eigenmode of Figure 3. Figure 5 illustrate the gain profile for the conventional CDRA designed with LTCC at the center frequency of 29.5 GHz.

### 2.2. New CDRA Design with a Grooved Superstrate

In the conventional CDRA design, the placement of the DR on top of the ground plane, where the radiation slot is located, is done manually, which leads to alignment precision issues that can easily impact the antenna gain and/or bandwidth. This issue can be compounded in an array setting. To tackle this issue, we propose a new LTCC stack, which consists of the stack of the conventional design on top of which a grooved superstrate, i.e., an additional LTCC layer with a hole, is added as shown in Figure 6. The groove is a precision laser milled during the LTCC fabrication process to generate a groove which is the exact size of the DR, within a 5 µm tolerance.

This new design allows for easy and precise alignment, however, as can be seen in Figure 7, it reduces the gain and deteriorates the front to back ration (FTBR) due to the excitation of the surface wave modes. Indeed, the gain is reduced by 0.46 dB, while the FTBR is degraded to 7.68 dB, down from 9.23 dB for the conventional CDRA stack. This behavior is to be expected, since for a conductor backed dielectric slab of thickness *d* and relative permittivity *ε_r_*, TM*_n_* and TE*_n_* modes can be excited, with cutoff frequencies given by Pozar [34]:(2)$$f_{c_{TM}}=\frac{nc}{2d\sqrt{\varepsilon_{r}-1}},\mathrm{for}\;n=0,1,2,\ldots$$
(3)$$f_{c_{TE}}=\frac{(2n-1)c}{4d\sqrt{\varepsilon_{r}-1}},\text{for}\;n=1,2,3,\ldots .$$

To eliminate all TE and TM modes having an index *n* ≥ 1, the thickness of the dielectric layer should be chosen such that TE_1_ is cut off at 31 GHz, the highest frequency in the considered band. This determines the maximum thickness that can be utilized as a superstrate. Here, we chose a superstrate thickness of *h_al_* = 0.03 mm corresponding to the thinnest available green sheet of LTCC Ferro A6M. This thickness ensures that all higher order modes are cut off while having minimum impact on the performance of the antenna. To eliminate the TM_0_ mode, which has no cutoff frequency, an additional ground plane is added to the top surface of the alignment superstrate and then the two ground planes (GND I and GND II) are connected with multiple vias as shown in Figure 8. Via fencing around the aperture is used as it prevents any radiation leakage from the sides in addition to ensuring that the two ground planes are connected. Consequently, the excitation efficiency is increased, and the total radiation efficiency is improved.

Figure 7 compares the gain profile of the CDRA when designed using the conventional stack, the grooved superstrate and the grounded grooved superstrate. As can be seen, the grounded grooved superstrate helps to eliminate the surface wave modes yielding an increase in gain by 1 dB over the CDRA with the grooved superstrate. When compared to the conventional CDRA, the grounded grooved superstrate yield a gain improvement of 0.53 dB, while enhancing the FTBR to 10.48 dB from 9.23 dB.

Figure 9 compares the realized gain variation over the considered frequency band of the proposed single element antenna for the convectional stack and the grounded and grooved superstrate.

Figure 10 illustrates the return loss for the proposed single element CDRA and compares the performance with the conventional method. It is shown that the bandwidth of 18.1% is achieved with the modified stack. The resonant frequency is designed to be centered at 29.48 GHz for the single element CDRA and there is a slight frequency shift between the conventional LTCC stack and the proposed modified LTCC stack.

## 3. Antenna Array Design

### 3.1. Geometric Arrangement of Array Elements

The proposed single element CDRA with grooved superstrate and dogbone coupling slot in the previous section is used to build a 4 × 4 rectangular array. The DRA elements are placed on top of the ground plane with a center-to-center distance of *S*, to be determined. The coupling slot dimensions are kept the same, while a corporate feeding network is added to feed each slot. Figure 11 illustrates the geometric arrangement of array elements with the feeding network underneath.

### 3.2. Inter-Element Spacing

One of the key parameters for designing the arrays of radiating elements is the inter-element spacing and the mutual coupling between adjacent elements that results from this spacing. This coupling will impact the radiation pattern, the appearance of grating lobes and the impedance matching. These characteristics cannot be predicted by array factor principals alone and must be taken into account by precise field analysis. While mutual coupling in DRAs is low compared to other types of arrays like patch arrays, they must still be taken into account in the design. To determine the optimal inter-element spacing for our array design, we carried out field simulations for a varying inter-element spacing, *S*, starting from *S* = 4.5 mm (0.38 λ_0_) to *S* = 10 mm (0.97 λ_0_) and computed the resulting gain as shown in Figure 12. A maximum gain of 15.68 dBi was achieved for *S* = 7 mm (*S* = 0.67 λ_0_). For *S* larger than 7 mm, grating lobes occur, resulting in a reduced gain.

### 3.3. Feeding Network Design

To achieve uniform excitation of all array elements, a corporate feeding network is chosen using the same LTCC stack. We used the inverted microstrip technology for simplicity and ease of implementation, though alternative lower loss guiding structures could be contemplated, i.e., SIW. The schematic of the proposed corporate feeding network is illustrated in Figure 12. The network is composed of multiple quarter-wave matched T-junctions. Each junction is made of a 50 Ω input line of width *W*_1_ = 0.7 mm connected to two 50 Ω output branches of the same width through a 35.5 Ω quarter wavelength transformer of width *W*_2_ = 1.4 mm and length *L*_2_ = 1.13 mm. A round chamfer of the 50 Ω output lines is introduced to improve bandwidth coverage and a chamfer radius of 0.7 *W*_1_ is found to yield optimal performance. Based on this design, the entire array with its feeding network fits into a square area the of size 25 × 25 mm^2^. However, to achieve a good FTBR value, i.e., better than 10 dB, while maintaining a compact size, the dimension of the ground plane must be optimized. An overall size of 46 × 46 mm^2^ is for the ground plane found to yield an FTBR of 11 dB and is selected for the fabrication. The array has been modeled and simulated with a full-wave simulator, Ansys HFSS. To ensure the highest accuracy when comparing it to measurements, a K-connector model was included in the simulation along with all dielectric and conductor losses. The simulation results of the designed array are presented in the next section and compared to the measured data.

## 4. Results and Discussion

The designed antenna array, including the feeding network and the dielectric resonator pucks, have been fabricated with the same LTCC stack [35]. Figure 13 shows pictures of the top and bottom sides of the fabricated antenna array. To measure the array’s performance without the effects of the connector, a TRL (Through, Reflect, Line) calibration technique was used where the K-connector and a short section of the inverted microstrip line were used to build the three TRL kit standards. To ensure an accurate comparison between simulation and measurement results, the simulated data were de-embedded from the K-connector’s plane to the same plane of the reflect standard of the TRL kit.

Figure 14 shows the simulated and measured results for the return loss of the array antenna. The measurement data have a slight frequency shift of ~0.2 GHz, compared to the simulation results, which may be due to minor imperfections during fabrication and/or a slight variation in the LTCC shrinkage during firing from the nominal values used in simulations. As shown, the measured 10 dB bandwidth of the antenna is 2.75 GHz, resulting in a fractional bandwidth of 9.81% around the resonant frequency of 28.52 GHz.

The simulated normalized gain profile of the proposed antenna array for the E- and H-planes at the center frequency is depicted in Figure 15. Figure 16 compares the simulated and measured realized gain pattern for the array at the measurement’s center frequency of 28.72 GHz. Overall, good agreement is observed at this frequency. The simulated 15.68 dBi and the measured 15.59 dBi gain values are virtually identical.

Additional comparisons between the simulated and measured radiation patterns are presented in Figure 17, simulations, and Figure 18, measurements, at different frequencies. The agreement between measurements and simulations is excellent around the center frequency with a gain difference of less than ±0.15 dB, but deteriorates slightly towards the band edges where the maximum gain difference reaches ±1.24 dB.

Figure 19 shows the variation of the measured efficiency of the proposed 4 × 4 CDRA array versus frequency at discrete points. A maximum efficiency of 88% has been achieved around 28.5 GHz and is at or above 85% between 27.5 and 30 GHz.

Table 2 summarizes the performance metrics of the presented DRA array and lists them alongside other DRA, microstrip patch antenna structures operating at mm-wave frequency bands in the literature. The proposed antenna array has a good gain and impedance bandwidth as well as a high efficiency, despite using a microstrip-type feeding network. Unlike all the other works, the proposed array is entirely fabricated in LTCC technology and used the novel grounded and grooved superstrate to achieve ease fabrication and alignment with improved gain and reduced FTBR.

## 5. Conclusions

In this work, a mm-wave cylindrical dielectric resonant antenna (CDRA) array is presented using a new LTCC stack that ensures precision and ease of alignment. The array antenna consists of 16 single element cylindrical DR antenna (CDRA) operating at 28 GHz for the 5G applications. The array is fed by an inverted microstrip corporate feeding network. Both the feeding network and the CDRA array are fabricated in the same LTCC process, ensuring cost effectiveness and high precision, thanks to the addition of a grounded and grooved superstrate. Measurement results show an impedance bandwidth of 9.81% around 28.52 GHz with a maximum gain of 15.68 dBi and an efficiency of 88%. These results are in excellent agreement with simulations and the slight discrepancies between the two are attributable to inherent variability that can be expected in a single run fabrication using LTCC due mainly to small shrinkage factor variation. With its compact size and the precision of its fabrication process, the proposed antenna is a good candidate for mm-wave applications.

## Figures and Tables

**Figure 1 sensors-21-03801-f001:**
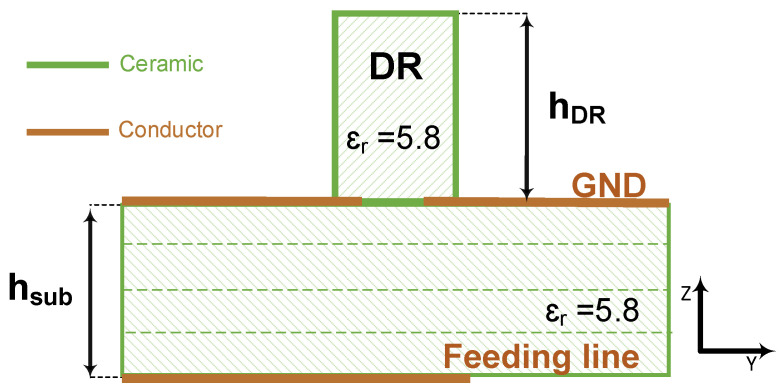
The schematic stack for single element CDRA.

**Figure 2 sensors-21-03801-f002:**
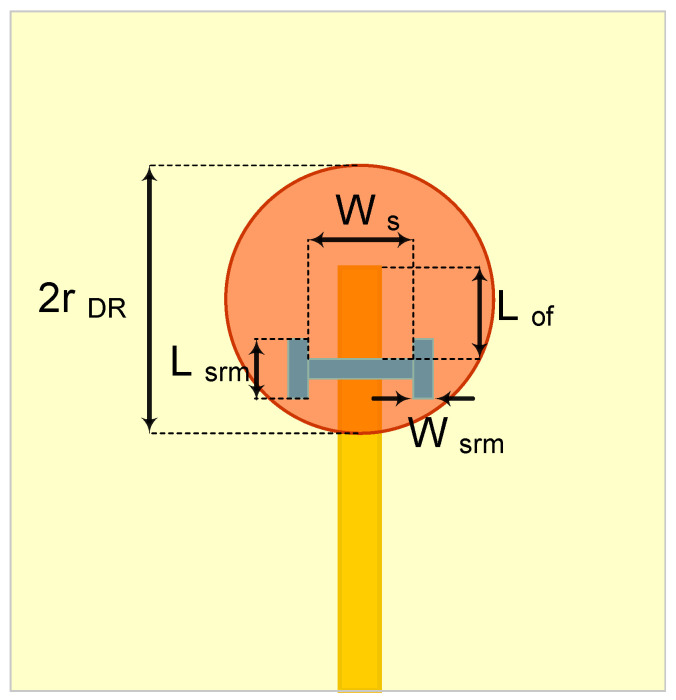
The schematic of the proposed single element CDRA (top view).

**Figure 3 sensors-21-03801-f003:**
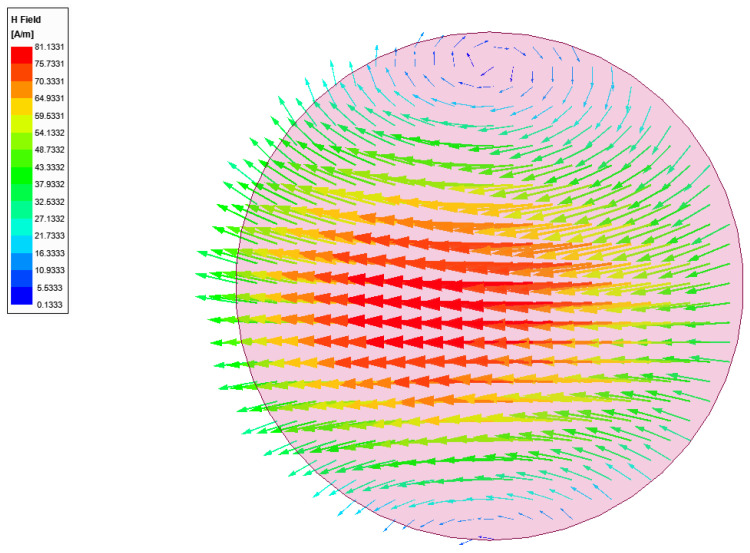
Magnetic field distribution of the resonant HEM_11δ_ mode in the proposed single element CDRA obtained from an eigenmode solution.

**Figure 4 sensors-21-03801-f004:**
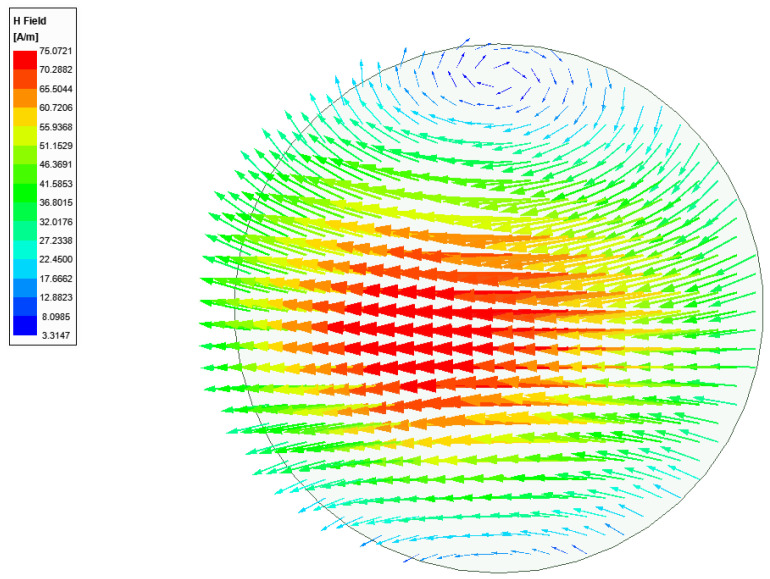
Excited near-HEM_11δ_ mode with the designed feeding structure of the CDRA.

**Figure 5 sensors-21-03801-f005:**
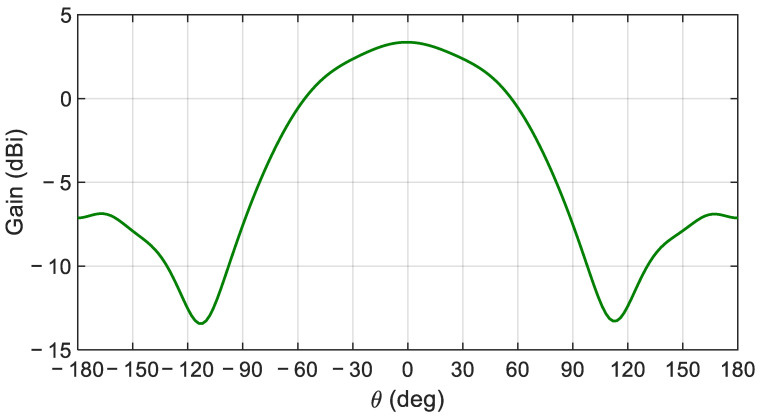
Simulated gain profile for the conventional CDRA design in LTCC at 29.5 GHz.

**Figure 6 sensors-21-03801-f006:**
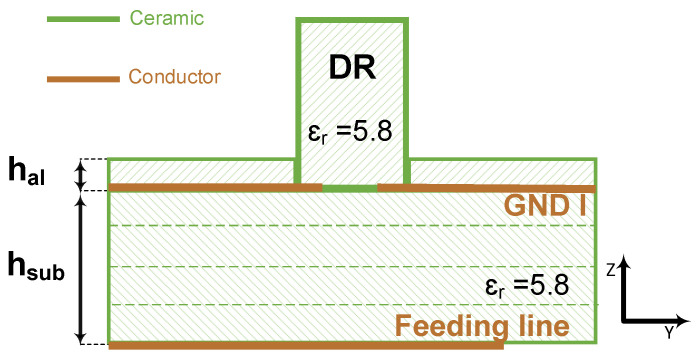
Schematic of the proposed stack with a grooved superstrate for the CDRA.

**Figure 7 sensors-21-03801-f007:**
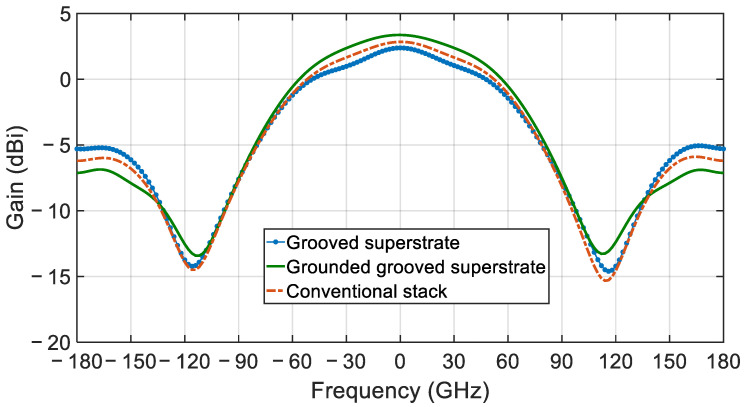
Comparison of the simulated gain profile for the single element CDRA for conventional, grooved superstrate and the grounded grooved superstrate stack.

**Figure 8 sensors-21-03801-f008:**
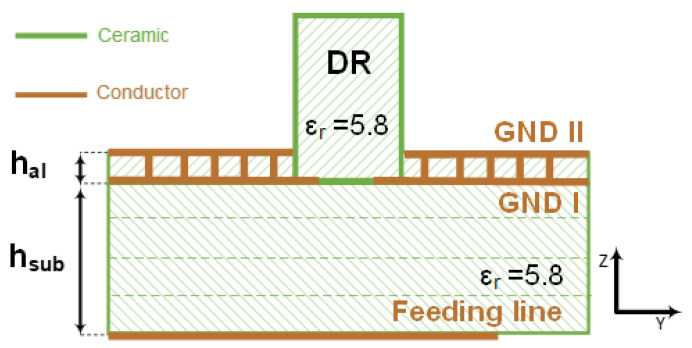
Schematic of the proposed stack with grounded and grooved superstrate for the CDRA.

**Figure 9 sensors-21-03801-f009:**
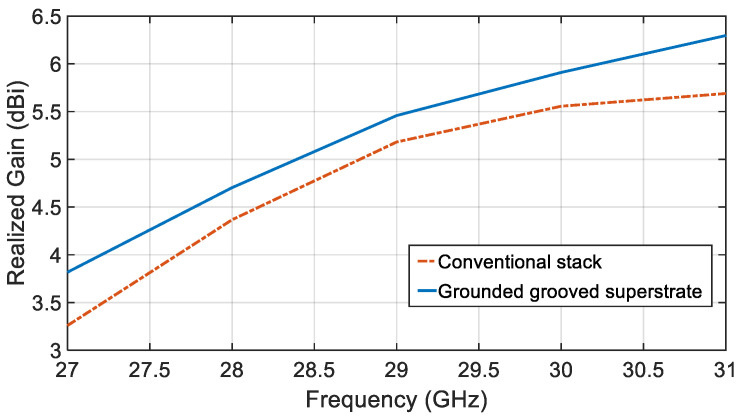
Realized gain profile for the single element CDRA for the two proposed stack over the operational frequency.

**Figure 10 sensors-21-03801-f010:**
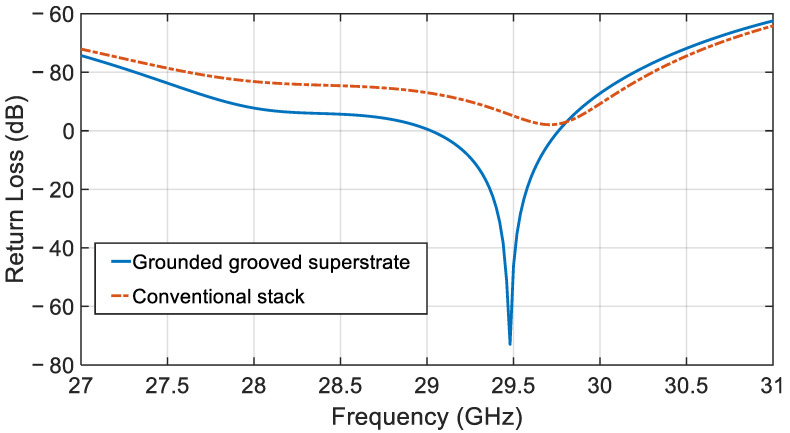
Return loss for the single element CDRA.

**Figure 11 sensors-21-03801-f011:**
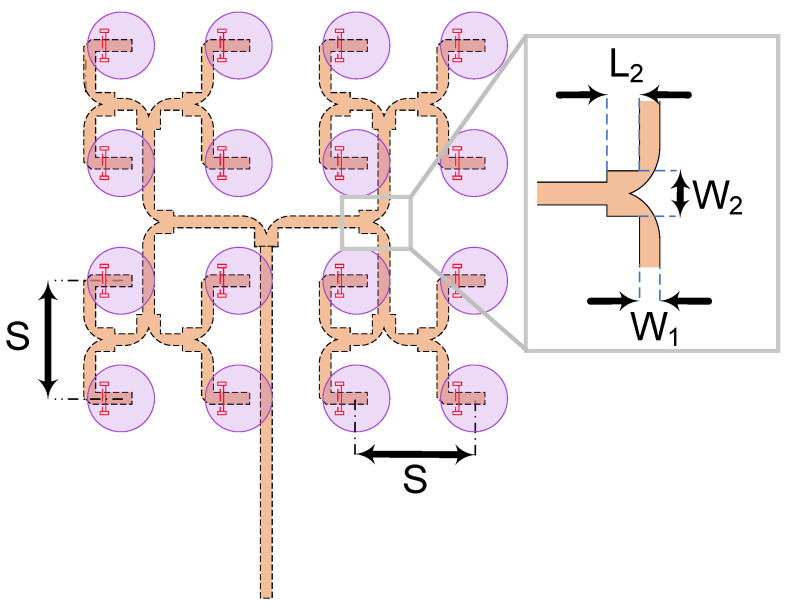
Feeding network schematic (ground plane hidden).

**Figure 12 sensors-21-03801-f012:**
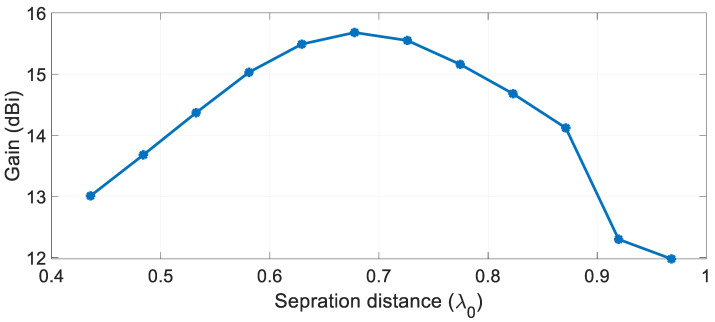
The gain profile variation for different inter-element spacing.

**Figure 13 sensors-21-03801-f013:**
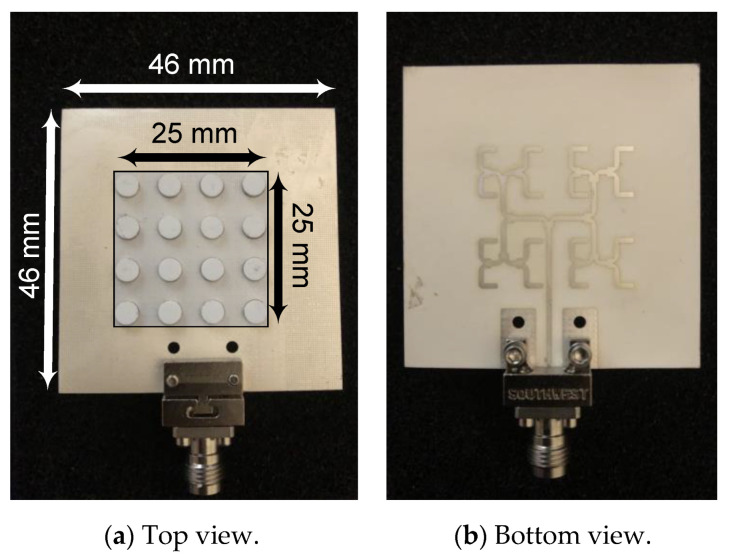
Fabricated DRA array antenna.

**Figure 14 sensors-21-03801-f014:**
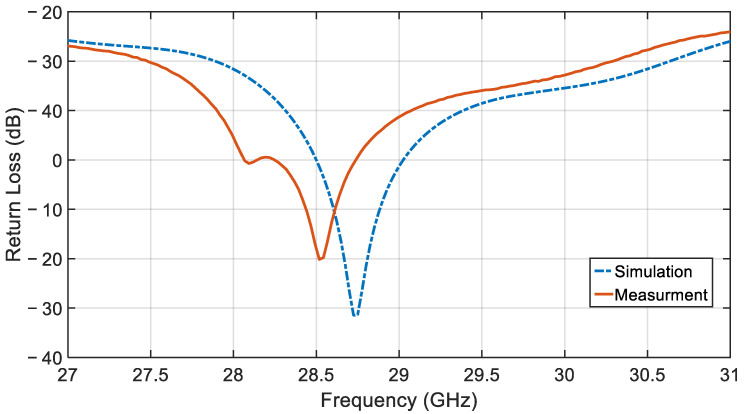
Return loss versus frequency of the proposed antenna array.

**Figure 15 sensors-21-03801-f015:**
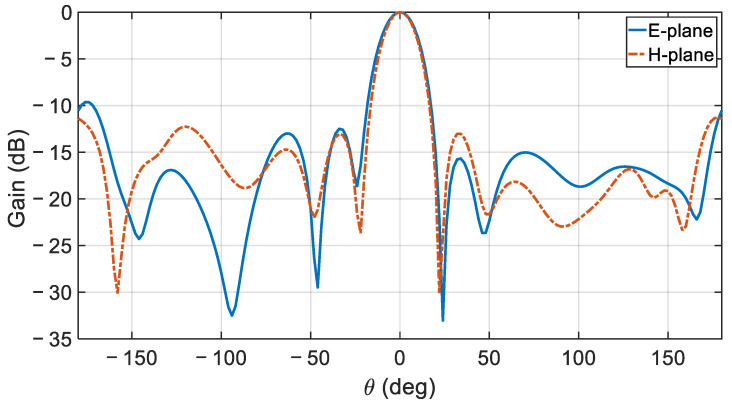
E- and H-plane radiation pattern at 28.52 GHz for the proposed CDRA array.

**Figure 16 sensors-21-03801-f016:**
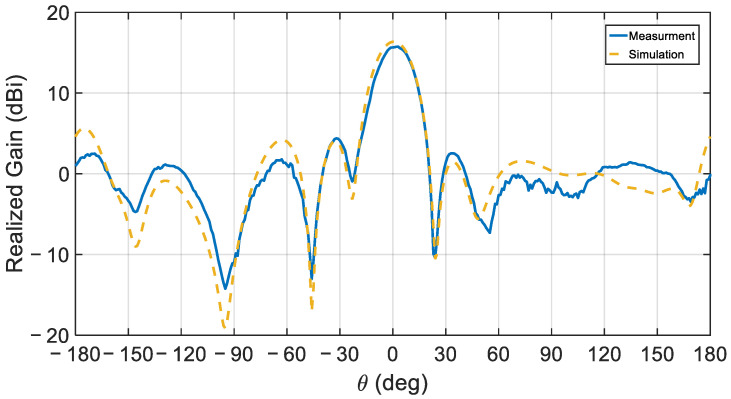
The simulated and measured radiation pattern of the CDRA array at 28.72 GHz.

**Figure 17 sensors-21-03801-f017:**
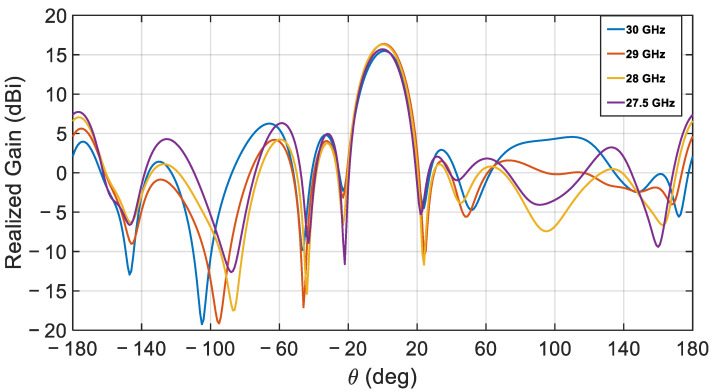
Simulated realized gain for the proposed CDRA array.

**Figure 18 sensors-21-03801-f018:**
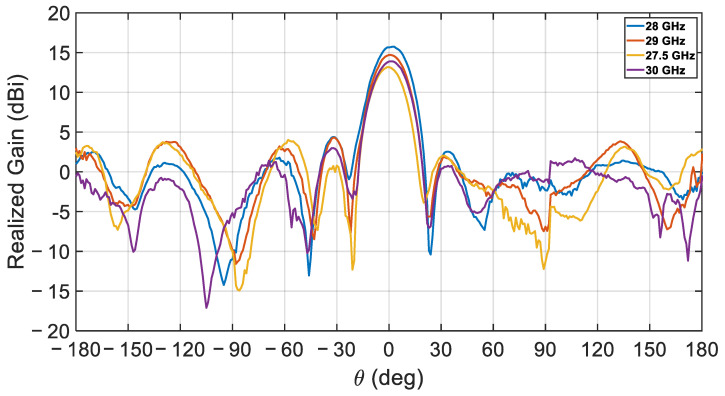
Measured realized gain for the proposed CDRA array.

**Figure 19 sensors-21-03801-f019:**
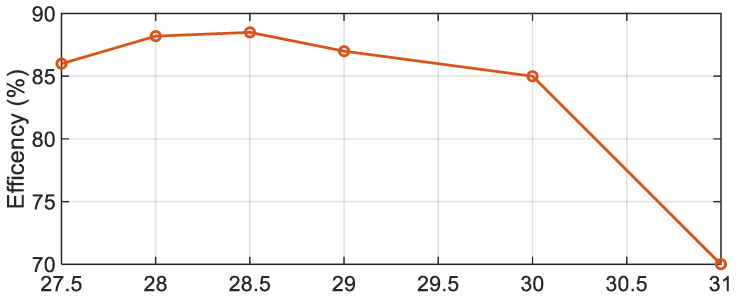
Measured radiation efficiency for the proposed CDRA antenna array.

**Table 1 sensors-21-03801-t001:** Design parameters of the proposed single element CDRA.

Parameters	Value (mm)	Description
*Lsrm*	0.5	Slot arm length
*Wsrm*	0.2	Slot width
*W_s_*	1.75	Slot length
*Lo f*	1	Offset length
*rDR*	1.95	Dielectric radius
*hDR*	1.45	Dielectric height
*hsub*	0.50	Substrate thickness
*w_f_*	1.75	Microstrip width

**Table 2 sensors-21-03801-t002:** Summary of performance of the proposed array alongside some of other reported antenna arrays in the literature.

Ref.		Center Frequency (GHz)	Bandwidth (%)	Gain (dBi)	Efficiency (%)	Elements
This work		28.7	9.81	15.68	88	16
DRA	[36]	36.7	6.81	12	91	12
	[3]	25.7	1.16	16.3	74	4
	[37]	29	3.3	7	80	4
	[38]	36	5.5	21.6	89	64
Patch	[10]	28	6.3	21	-	42
	[39]	29	6	13	75	8
	[40]	28	2.6	11	70	-

## Data Availability

Not applicable.

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
