# Peer review of "LTCC-Integrated Dielectric Resonant Antenna Array for 5G Applications"

_sensors, 2021, doi:10.3390/s21113801_

Round 1

Reviewer 1 Report

  1. In the section of element design, parameter discussion is not sufficient, in fact basically not involved.
  2. When forming array, influences of element distance and mutual coupling are not presented.

Author Response

  1. Additional detailed descriptions of the parameters and how to obtain their values have been added to section 2.1 of the manuscript.
  2. The manuscript has been revised and a new section has been added to detail how inter-element spacing is selected.

Reviewer 2 Report

This paper has been examined in detail. Overall, this work was conducted on a well-investigated subject, and thus, has got an extensive body of available literature. Hence, the authors must conduct a more comprehensive background study on the general area of mmWave 5G DRAs, including their various implementation methods (e.g., 3D printed, flexible antennas, etc). In this regard, the Introduction section, as well as Table 2 should be extended considerably, to include a thorough qualitative comparison. It is also recommended to include a brief theory behind the DRA concept, so the overall structure and its figures of merit could be better understood. The English language should undergo a slight review and improvement as well, to ensure the error-free and smooth presentation of the contents.

Author Response

  1. The introduction has been revised and has been updated and additional references on flexible and 3D printed mm-wave antennas have been added. Table 2 has been extended to include DRA and non-DRA antennas.
  2. A very short discussion of the theory behind the DRA concept is added to section 2 and a reference to a book that discusses the well-established theory in detail is added.
  3. The manuscript has been revised and errors were corrected.

Reviewer 3 Report

Dear authors, my comments and suggestion I'm offering you for improving your work.

As a general comment, the paper is well structured and the topic is presented and discussed in a proper manner.

row 32: PMC acronym not defined, please define.

row 57: please add acronym after Low Temperature Cofired Ceramics, otherwise LTCC is not properly reffered

row 79: " ... but and ..." seems not grammarly corrected

row 89: " .. the same procedure ..." is not clear "same to what reffered procedure

Figure 2 and Equation 1: choose radius or diameter, but please coherent between figure and equation (d_DR or r_DR)

row 109 and row 119: why using two different full wave simulator? That may be misleading since CST and HFSS use different numerical techniques.

Figure 3 can be omitted, also it doesn't seem original from the simulator. Figure 4 is the same physic representation.

Figure 5 and others: please add "simulated" gain profile.

row 146: "with a whole", word is missing somewhere

row 153: typo "as can been in"

row 154: FTBR acronym not defined

Figure 7 can be omitted, it is included in Figure 9

A general comment about the array design. Are you aware that your design it's working only in TX mode?

After Figure 15 the Figure numbers are wrong, please correct them.

Table 2, it reports a not correct comparison, since other designs work at different frequencies and/or consist of a smaller/bigger number of antenna elements. What are you comparing? What do you want to highlight?

Author Response

row 32: PMC acronym not defined, please define.

Line 34 has been updated with the ancronym.

row 57: please add acronym after Low Temperature Cofired Ceramics, otherwise LTCC is not properly reffered

The manuscript has been updated on line 59  based on this comment.

row 79: " ... but and ..." seems not grammarly corrected

The manuscript has been updated on line 81  based on this comment.

row 89: " .. the same procedure ..." is not clear "same to what reffered procedure

The manuscript has been updated on line 94  based on this comment.

Figure 2 and Equation 1: choose radius or diameter, but please coherent between figure and equation (d_DR or r_DR)

Figure 2 has been updated.

row 109 and row 119: why using two different full wave simulator? That may be misleading since CST and HFSS use different numerical techniques.

This has been resolved. 

Figure 3 can be omitted, also it doesn't seem original from the simulator. Figure 4 is the same physic representation.

Figure 3 has been removed.

Figure 5 and others: please add "simulated" gain profile.

Figure 5 caption is revised.

row 146: "with a whole", word is missing somewhere

line 154 Has been revised.

row 153: typo "as can been in"

line 161 has been revised.

row 154: FTBR acronym not defined

Line 162 has been updated with defined acronym of FTBR

Figure 7 can be omitted, it is included in Figure 9

The figure is removed.

A general comment about the array design. Are you aware that your design it's working only in TX mode?

We do not fully understand the reviewer’s comment/question. The array is made of all isotropic materials and is fully reciprocal. Therefore, its radiation pattern is the same for transmit and receive.

After Figure 15 the Figure numbers are wrong, please correct them.

Figure numbering is corrected.

Table 2, it reports a not correct comparison, since other designs work at different frequencies and/or consist of a smaller/bigger number of antenna elements. What are you comparing? What do you want to highlight?

Since mm-wave 5G antennas are relatively recent, there are not many repetitive designs in the open literature, i.e., there are no identical designs only various alternative designs at different frequency with different number of elements. Therefore, Table 2 provides more of a summarized overview of available options and their performances and lists our own work as one of the options. We removed the term ‘comparison’ to avoid any ambiguity.

Round 2

Reviewer 1 Report

In view of the innovation and depth of research, it is suggested to reduce the length of this paper.

Author Response

The authors believe that the length of the manuscript is appropriate and could not identify any superfluous material that could be eliminated without negatively impacting the overall cohesion and readability of the paper.

Reviewer 2 Report

The revised version of the paper has been examined in detail. Overall, the authors have addressed the major points raised previously and have incorporated new materials into the body of the current version. It is acceptable to proceed with its publication. A final English revision should be conducted though, in order to ensure the grammatical and article error free of the text.

Author Response

The manuscript has been revised and some errors were corrected.